# An examination of gender imbalance in Scottish adolescents' vocational interests

Laurence Lasselle[1]*, Stijn Schelfhout[2,3], Lot Fonteyne[4], Graham Kirby[5], Ian Smith[6], Wouter Duyck[3]

1 School of Management, University of St Andrews, St Andrews, United Kingdom, 2 Department of Work, Organisation and Society, Ghent University, Ghent, Belgium, 3 Department of Experimental Psychology, Ghent University, Ghent, Belgium, 4 Department of Educational Policy, Ghent University, Ghent, Belgium, 5 School of Computer Science, University of St Andrews, St Andrews, United Kingdom, 6 School of Economics & Finance, University of St Andrews, St Andrews, United Kingdom

* laurence.lasselle@st-andrews.ac.uk

**Data Availability Statement:** Data cannot be shared publicly because of privacy regulations on pupils' data and the information given to the participants. This research was granted ethical approval by the University of St Andrews Teaching

## Abstract

This paper documents Scottish adolescents' vocational interest types. Our research is based on the responses of 1,306 pupils from 18 secondary schools to an empirically verified online interest inventory test. Our results are threefold. First, the structural validity of the test with the Scottish sample is confirmed by evaluating the underlying circumplex structure of Holland's RIASEC vocational interests. Second, gender distribution along the six primary vocational interest dimensions is consistent with the research literature: young men scoring higher on the Realistic vocational interest and young women scoring higher on the Social dimension. Finally, we observe that across dimensions, vocational interests of young women are less diverse than those of young men. We discuss how these dissimilarities could lead to differences in education choice and career decision-making.

## Introduction

Personal vocational interests predict educational choice [1] and career pathways [2]. As such, adolescents will benefit from exploring these interests when deciding their subject choices, or whether to enrol in higher education or immediately move into employment. This exploration should lead them to find educational or work environments which match their interests, allowing them to flourish [3].

Vocational interests have multiple characteristics, including that of being relatively stable from early adolescence to middle adulthood ([4,5]). This stability can explain why guidance teachers at secondary schools or career-counselling services at colleges and universities often make use of them to provide information, assistance and advice to pupils and students in terms of selection of academic subjects, career opportunities and/or access to professions. It can also explain the motivation to describe the labour market and its long-term trends according to workers' vocational interests ([6–8]). Trends then signal future employment opportunities which in turn could facilitate the planning of pupils' careers.

Vocational interests are assessed through responses to interest inventories, often derived from the Holland RIASEC theory [2], one of the most influential models of vocational

and Research Ethics Committee (UTREC), with reference MN10343. Researchers who meet the criteria for access to strictly confidential data may make a request via utrec@st-andrews.ac.uk quoting MN10343. Please note that the participants were clearly informed that only the researchers will have access to the data, which will be kept strictly confidential.

**Funding:** The author(s) received no specific funding for this work.

**Competing interests:** The authors have declared that no competing interests exist.

interests. The central tenet of this theory is straightforward. People tend to thrive in studying and working environments that fit their interests. Holland used six primary dimensions, classifying both people's interests and people's environments on the same template, allowing for commensurate measurement: Realistic (R), Investigative (I), Artistic (A), Social (S), Enterprising (E) and Conventional (C). The Holland model has been extensively verified on empirical data and has led to the development of multiple interest inventory tests, linking interests, academic subjects and careers around the world ([9–11]).

Our paper explores the implications of the diversity of interest profiles among pupils and discusses how this information could be relevant for guiding teachers and policy makers. To begin with, we examine responses of Scottish adolescents to a validated and contemporary interest inventory tool [11]. This instrument allows us to establish a vocational interest profile for each adolescent. We then study the gender diversity of these profiles using the well-known Simpson's diversity index adapted to an educational framework by [12]. Our specific purpose is to provide the first documentation of the range of vocational interests among young Scots.

To facilitate the discussion, we limit the presentation of this first documentation to Holland's one- and two-dimensional interest types. Our approach is quite typical of this kind of research. The studies mentioned previously for the characterisation of the labour market used Holland's one-dimensional types. Vocational interests corresponding to each academic subject are determined by one-dimensional types ([13,14]). Our tool is derived from that of [11], in which all vocational interest profiles of study programmes are defined by a two-dimensional type. In other words, in our presentation, vocational interests of the adolescents are characterised by the one or two most dominant (i.e., highest scoring) Holland interest dimensions (R, I, A, S, E or C). We compare and contrast the distribution (i.e., absolute numbers and relative proportion) of the dimension scores between the young women and men participating.

## Background

### Holland theory and RIASEC test

In his theoretical work, Holland proposes a model that links the vocational interests of individuals to their working or studying environment [2]. For example, people with primarily an Investigative interest, scoring highest on the I dimension, would be more likely to work or study in an Investigative environment, for example in a scientific discipline such as physics or chemistry. Holland's six interest dimensions may be visualised on a hexagon arranged in a clockwise order, reflecting the relations between them, as shown in Fig 1.

The shorter the distance between two different dimensions, the greater their similarity. Hence, the distance between R and I is the shortest. As a pair, the types are most similar and are said to be adjustment types. On the contrary, the distance between R and S is the greatest, and they are said to be opposite types. R and A are seen as moderately related and are said to be alternate types. A high similarity between person's profile and the work (or study) environment profile indicates a good fit. For instance, students with Realistic profiles in Realistic environments are perfect fits. The same students in Investigative or Conventional environments represent moderate fits. If they were in a Social environment, the level of fit would be low.

In practice, the measurement of people's vocational interest dimensions rests on taking an interest inventory test (also called a RIASEC test). The test gives a score to each dimension and this constitutes the full RIASEC profile. Holland advocated the use of this full profile where possible, in order to optimally match people with environments [2]. However, as we shall see, the use of full profiles would be meaningless in our interest diversity investigation and subsequent gender difference examination.

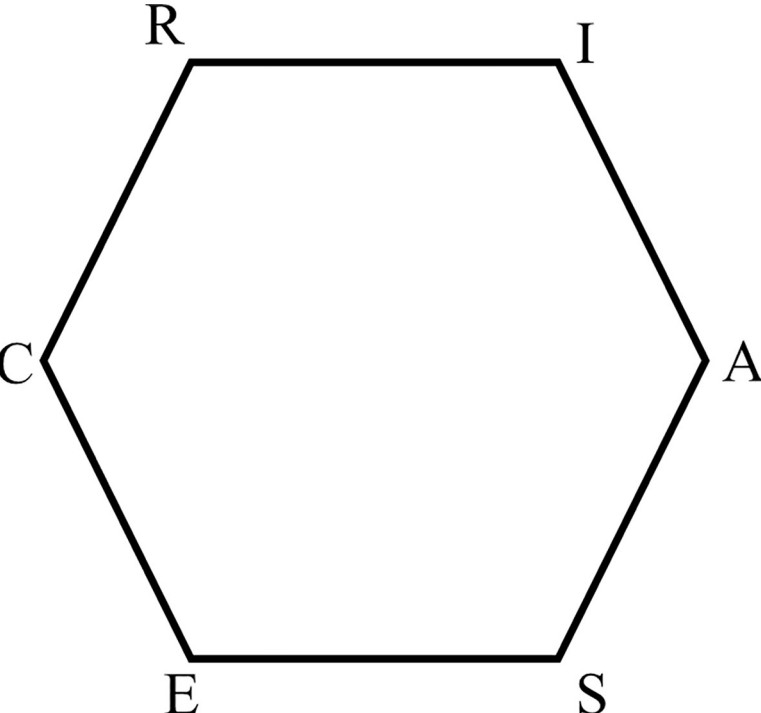

**Fig 1. RIASEC model of vocational interest types and environments (from [15]).**

Literature on vocational interests has highlighted gender differences in terms of profiles both within and across countries ([5,16–19]). Realistic and Investigative interests are often strong for men and Artistic, Social and Conventional interests for women. Stewart-Williams and Halsey provide an informative discussion of the many factors at play when it comes to explaining the differences between men and women in a number of fields in STEM (science, technology, engineering and mathematics), including the greater male variability in a number of traits (vocational interests being one of the traits) [19]. Taken together, previous research evidences the strength of the vocational interest circular representation (i.e., hexagon) across countries. Variation in cultural and/or economic settings may explain differences in the structural validity of the inventory test and its replication across the globe ([18] in particular).

## Diversity

Our diversity study is based on the concepts of richness and evenness presented by Kelly in an educational framework [12]. In our context, we use them to evaluate the heterogeneity of vocational interests among our participants. The number of vocational interest profile types is called 'richness' and the relative frequency of these types is called 'evenness'. Let us take an example to illustrate both measures. The distribution of RIASEC one-dimensional interest types among 100 young women may be 10, 5, 40, 40, 2, 3 respectively. In other words, 10 young women have R for one-dimensional interest type, 5 have I, etc. The distribution among 100 young men is 15, 25, 14, 26, 13, 7. Both groups have the same number of participants (100) and the same richness (6). However, the young men group has greater evenness: the participants are more evenly distributed across the six one-dimensional types. In other words, their profiles are more diverse than the young women's profiles. For the present study, we calculate evenness using the inverse Simpson index (cf. Appendix A for the derivation). This is often

used in ecology and economics to assess the degree of concentration of species in nature or firms in an industry.

## Data and methodology

### Participants, schools and online questionnaire

Our data were obtained from Scottish adolescents' responses to an online RIASEC test collected between November 2017 and October 2019 (we give more information about the test in Appendix B). This test assessed the vocational interests of 1,639 pupils from two independent (private) schools and sixteen publicly funded secondary schools. These pupils were in S5/Year 12 and in S6/Year 13, i.e., the two final years of secondary education in Scotland. As in [11], we discarded submissions with more than five percent of missing values (nine items or more), leaving us with the responses of 1,306 pupils. Of these respondents, 704 participants (54%) were young women, 577 (44%) young men, and 25 (2%) preferred not to disclose their gender.

All the schools offered their pupils the opportunity to take part in the research. Pupils took the test during a session devoted to Information Technology, Higher Education/Further Education studies or the work environment. A few pupils took the test at home due to technical issues during their class hours. 80% of the respondents took the test when in S5. To avoid duplicate entries, pupils who had already taken the test in S5 were not invited in S6. The number of participants per school varied greatly, from 9 to 230, due to school size and location (e.g., the school with the smallest number of participants is located on an island). The list of schools is available in the Acknowledgements section.

### Online questionnaire design

The online questionnaire is based on the survey developed by [11], with the participant's personal self-declared information adapted to the Scottish context. Responses were anonymous; participants accessed the questionnaire via a pre-generated access code from which their school could be deduced, but no other personal information.

At the beginning of the survey, participants were asked to self-declare their gender and their SQA Higher subjects. (SQA Highers are Scottish qualifications that can lead to university, further study, training or work. They are selected by the pupils and usually taken in their fifth year of secondary school.) The participants then accessed the RIASEC questionnaire, which was divided into two sections, containing statements (the so-called 'items') regarding activities and occupations respectively. For each statement, they were asked to select one of *Like*, *Don't like* or *Prefer not to answer*. As such, there were no mandatory questions. Unknown to the respondents, each activity and occupation is associated with one of the RIASEC dimensions. For instance, 'Managing a database' is an R activity, while 'Business economist' is a C occupation. The mapping of statements to interest dimensions is that of [11].

### Research methods

Our documentation of Scottish adolescents' vocational interests is based on their responses to the RIASEC questionnaire. To proceed, we first check the structural validity of the application of a Flemish inventory interest tool to Scottish data. Validity is always a function of instrument and specific sample ([20,21]). The tool developed by [11] has been designed to suit the Flemish education system while also confirming the theoretical circular structure proposed by [2]. However, we should not consider cross-cultural replication of this Flemish instrument as a given in the Scottish context. We must therefore first evaluate whether the RIASEC pattern of our sample matches the circular structure proposed by Holland (recall Fig 1). Following a

positive validation, we can then compare and contrast the distribution of interest types between the young women and men participating.

**Estimation of the vocational interest type.** For each of the RIASEC dimensions, the algorithm calculates the percentage of *Like* responses for activities and occupations associated with that dimension. (Each response has three possibilities: *Like*, *Don't like* and *Prefer not to answer*. By combining the latter two, each response becomes binary.) The RIASEC full profile of each participant can then be constructed from the percentage score of each dimension. The RIASEC profile of group samples (i.e., total, young women, young men) is computed from the average scores on each of the six dimensions. On completion of the questionnaire by the respondent, the tool displays a three-dimensional interest type. This is a three-letter code, identifying the three dimensions with the highest percentages, with a brief explanation as immediate feedback. The first letter is the one-dimensional interest type, the second is the dimension with the second highest percentage and the third that with the third highest percentage.

**Structural validity.** The structural validity of the tool in our context has to be confirmed by evaluating the underlying circumplex structure of the RIASEC model To do so, we use both a confirmatory factor analysis (CFA, [22]) and a randomization test of hypothesized order relations (9 with, [23,24]). A CFA (parametric) provides a full analysis of the variance–covariance matrix of the RIASEC data, while a RTOR (non-parametric) focuses exclusively on evaluating the possibility of a circular structure [17]. For the RTOR, Holland's theory allows us to make 72 order predictions in terms of correlations between interest types. A circular structure predicts correlations between adjacent dimensions (e.g., R and I) to be higher than correlations between alternate dimensions (e.g., R and A) and opposing dimensions (e.g., R and S). Specifically, RTOR evaluates the percentage of predictions met based on the available data [25]. As literature is still equivocal on which method is preferred (i.e., CFA or RTOR) to evaluate circular fit [17], we report both methods. In other words, given the importance of gender differences, we conduct a sequence of three circumplex analyses using first CFA and then RTOR: for the total sample, and also for young women and men separately. In all cases, we consider adolescents' full RIASEC profiles [17].

**Gender differences.** After structural validation, we examine the gender disparity of profiles among the participants in two different ways. First, we consider the group profiles of both young women and men to determine differences in terms of primary RIASEC dimensions for each gender group. Second, we opt to reduce the number of dimensions and resort to high-point coding, as is common practice in both literature ([26]) and practical application of RIASEC instruments. We classify each RIASEC profile as a one-dimensional interest type and a two-dimensional interest type based on the highest scoring dimensions. The one-dimensional interest type of a pupil is the dimension (represented by the appropriate RIASEC letter) in which they obtain the highest score following their responses to the RIASEC questionnaire. The two-dimensional interest type consists of the two dimensions (again represented by the appropriate RIASEC letters) in which a pupil obtains their highest scores. The order of the two letters is interchangeable in our examination. For instance, let us consider a group composed of two young women. The first respondent has SECRIA as a RIASEC profile. Her one-dimensional interest type is S (out of 6 possibilities) and her two-dimensional interest type is SE (out of 15 possibilities). The second participant has ESRIAC as a RIASEC profile. Her one-dimensional interest type is E and her two-dimensional interest type is ES. We will say that the group has two one-dimensional interest types (S and E), and a single two-dimensional interest type (SE or ES). We also conducted our examination with letter ordering preserved. The results are robust to this alternative analysis.

For each gender group, we pool all respondents with the same vocational interest profile in each interest type. We then calculate their relative frequency. This allows us to inspect the diversity of vocational interest profiles within each type among both gender groups based on the inverse Simpson's index. This index takes values between 0 and 1. When this index equals 0, all profiles are identical, i.e. there is no diversity in terms of vocational interests. Diversity increases as values move away from 0 and reaches its maximum when the index equals 1. For instance, in the case of the one-dimensional interest type, a value of 0 indicates that all participants share the same vocational interest, captured by the letter of the dimension. The sample of participants' profiles has no diversity. On the contrary, a value of 1 implies that respondents' profiles are distributed evenly across the six possible one-dimensional interest types. The sample would then be the most diverse. (Kelly computes various diversity indexes in an educational context [12]. For ease of presentation, we selected the Simpson's index in this paper. The derivation of this index used is available in Appendix A. Our results are consistent across the indices.)

The use of Simpson's index allows us to enrich the discussion on interest types. Recall that interest types are indicative of profession destinations. If we report less diversity in vocational interest types, then participants could also display less diversity in the professions they will eventually enter.

**Limitations of our approach.**   Our paper documents the vocational interests of young adolescents residing in Scotland and as such has two limitations we ought to highlight at this stage. First, our diversity analysis is not taking into account fully the dominance of the dimension. The score determines the type, but the analysis only rests on the proportion of respondents within each dimension. For example, two pupils who both have the one-dimensional interest type R, with respective score of 80% and 35% are treated similarly. Second, we do not consider the popular three-dimensional interest type.

The strength of our approach lies in its simplicity. First, for purposes of study advice or work orientation, it is always the relative order of interests that steers decisions. In the above example, for both scores, the dominant dimension is identical. So both pupils will very likely make the same decision. Second, the use of one- and two-dimensional vocational interest profiles allows us to show the relevance of vocational interests in our context of diversity and why they should be studied further. Although Holland does advocate the use of the full profile, the approach of high point coding towards the use of the highest scoring dimensions (up to three dimensions) is widely accepted in the literature as a practical, more user-friendly alternative ([2,11]). Note that the three-dimensional interest type (with non-interchangeable letters) implies 120 possible combinations of three-letter codes. The distribution of each of the two sub-groups would be meaningless in our case, as the total number of respondents does not exceed 1,300. An even distribution of young women among the three-letter codes would gather less than 6 pupils per code and that of young men less than 5.

**Ethics statement.**   The authors used primary data. Their research was granted ethical approval by the University of St Andrews Teaching and Research Ethics Committee (with reference MN10343). It was carried out in accordance with the requirements of this committee regarding online informed consent for the participants, the opt-out procedure for parental consent and data availability.

Prior to filling out the online surveys, pupils had access to the participant information sheet. This included information on the topic of the study, their voluntary participation in the study, the confidentiality of their participation, data storage and destruction and how the results would be published. In particular, pupils were assured that they could omit any questions and that only the researchers would have access to the data, which would be kept strictly confidential.

They were also clearly informed that their participation would not affect their progression to a HE institution, in particular the University of St Andrews. At the bottom of the online participation page, they indicated their willingness to participate in the study by clicking on the box starting the questionnaire. If they did not want to participate, they were informed that they simply had to close the page. The participants were S5/Year 12 and S6/Year 13 students. Some of the S5 pupils could be less than 16 years old at the time they completed the questionnaire. As the age of full legal capacity is 16 in Scotland, parents or guardians whose children were younger than 16 years old were asked to inform the school office, using an opt-out form, if they did not want their child to be involved in the research.

## Results

### Structural validity

**Descriptive statistics and internal consistencies.** Table 1 assembles the number of items and the internal consistency of the six dimensions (R, I, A, S, E and C) for the overall survey and each of its components, i.e., Activities and Occupations. Overall, the Cronbach's alphas range from 0.86 (Investigative and Conventional) to 0.89 (Enterprising). The Cronbach's alphas in Activities are slightly higher than those in Occupations in half of the cases. The former range from 0.76 (Investigative) to 0.84 (Social), the latter from 0.71 (Conventional) to 0.82 (Enterprising). Although the values are slightly below than those of the Flemish benchmark, they are well above the acceptable value of 0.7 which indicates good consistency.

The correlations between the six dimensions are given by Table 2. All significant correlations are positive, in line again with the benchmark.

**Evaluation of circumplex structure.** The circular structure of the proposed RIASEC dimensions is evaluated using a CFA (CirCe package for R, [22,27]) and a RTOR (RANDALL package for R, cf. also http://tracey.faculty.asu.edu/computercov.html, [23,24]). This evaluation is done for the entire data set (1,306) and for the 577 young men and the 704 young women separately. Results are presented in Table 3. The circumplex fit is illustrated by Fig 2.

Fig 2 shows the circular structure for both the complete dataset as well as for young women and men separately. The best fitting models for the present study are models with unequal spacing of scale locations (varying distance between locations), but equal scale communalities (i.e., locations are positioned on the circle). As the $\chi^2$ test is significant for all three models but oversensitive to minor deviations of a good model fit, we report a set of common fit indices to further evaluate circular model fit. The CFA fit indices from Table 3 report somewhat mixed results regarding circular model fit (cf. also Fig 2). Although most index values are of at least acceptable quality, the RMSEA values are too high, especially for young women. These results were also reported in the description of the original Flemish instrument [11]. However, Kenny and colleagues state that the RMSEA value can become artificially high for models with a small

**Table 1. Dimension descriptive statistics, internal consistencies and number of items.**

|   | Activities | | | | Occupations | | | | Total | | | |
|---|---|---|---|---|---|---|---|---|---|---|---|---|
|   | *N* items | *M* | *SD* | α | *N* items | *M* | *SD* | α | *N* items | *M* | *SD* | α |
| R | 14 | 24.90 | 25.10 | .81 | 8 | 21.14 | 24.76 | .73 | 22 | 22.69 | 22.76 | .87 |
| I | 15 | 30.98 | 23.17 | .76 | 14 | 19.03 | 21.46 | .80 | 29 | 25.19 | 20.55 | .86 |
| A | 13 | 38.59 | 26.18 | .78 | 13 | 25.50 | 23.23 | .79 | 26 | 20.92 | 21.68 | .87 |
| S | 18 | 37.47 | 26.83 | .84 | 10 | 27.29 | 25.01 | .77 | 28 | 21.01 | 21.09 | .88 |
| E | 13 | 32.71 | 26.59 | .80 | 11 | 21.49 | 24.77 | .82 | 24 | 27.53 | 23.97 | .89 |
| C | 14 | 24.90 | 25.10 | .82 | 9 | 15.82 | 20.37 | .71 | 23 | 21.34 | 21.32 | .86 |

Notes: M: Mean; SD: Standard Deviation; α: Cronbach's alpha; N: Number of items.

**Table 2. Interest type intercorrelations.**

|   | R | I | A | S | E | C |
|---|---|---|---|---|---|---|
| **R** | 1 | .403** | .278** | -.009 | .273** | .395** |
| **I** |   | 1 | .319** | .372** | .345** | .420** |
| **A** |   |   | 1 | .436** | .410** | .274** |
| **S** |   |   |   | 1 | .521** | .346** |
| **E** |   |   |   |   | 1 | .737** |
| **C** |   |   |   |   |   | 1 |

Note:

**$p < .01$ (2-tailed).

number of degrees of freedom (*df*) (i.e., *df* = 6 for the present study), and should therefore be interpreted with caution [28]. They therefore advise both researchers and reviewers not to dismiss models with higher RMSEA values and lower degrees of freedom without taking into account other sources of information. For this reason, we have considered RTOR analysis and several other empirical studies that evaluate circular fit in young individuals in our examination.

Table 4 presents the outcomes of RTOR. These outcomes seem to indicate a good model fit. About 80% of the order predictions are met, resulting in correspondence indices (CI) that are well above the international benchmarks (CI = 0.48) suggested by Table 4 presents the outcomes of RTOR. These outcomes seem to indicate a good model fit. About 80% of the order predictions are met, resulting in correspondence indices (CI) that are well above the international benchmarks (CI = 0.48) suggested by [29]. Consequently, hypothesis testing confirms the theorized circular fit, as the hypothesis tests reject the null hypothesis of a random ordering of the RIASEC types in favor of a circular ordering.

Several empirical studies, including [30] and [31], show evidence that the RIASEC circular structure is not yet fully developed in younger pupil populations. As the participants in the present study (16 to 18 years) are younger than those in the original study (18 to 19 years), minor deviations regarding circular fit can be expected.

As such, we suggest that the CFA and RTOR analyses combined provide reasonable evidence that the a priori theoretical circular RIASEC model has a sufficient fit with the data. The use of the Flemish SIMON instrument is therefore suited for cross-cultural application to a Scottish context.

## Gender differences

Our dataset is composed of 25 (2%) pupils with undisclosed gender, 577 (44%) young men and 704 (54%) young women. According to Table 5, young men scored higher on the Realistic, Investigative, Enterprising and Conventional dimensions compared to young women. In

**Table 3. Overview circumplex goodness of fit indices.**

| Groups | $\chi^2$ (df) | RMSEA | SRMR | AGFI | CFI | df | P |
|---|---|---|---|---|---|---|---|
| **Young men** | 1409.28 (15)*** | .12 | 0.05 | 0.91 | 0.96 | 8 | 13 |
| **Young women** | 1245.57 (15)*** | .14 | 0.06 | 0.87 | 0.91 | 8 | 13 |
| **All pupils** | 3438.91 (15)*** | .11 | 0.05 | 0.91 | 0.96 | 6 | 15 |

Notes. $\chi^2$ (df) = chi-squared test with degrees of freedom

*** $p < .001$; RMSEA = Root Mean Square Error of Approximation; SRMR = Standardised Root Mean Square Residual; AGFI = Adjusted Goodness-of-Fit Statistic;

CFI = Bentler Comparative Fit Index; *df* = circumplex degrees of freedom; P = circumplex number of Parameters.

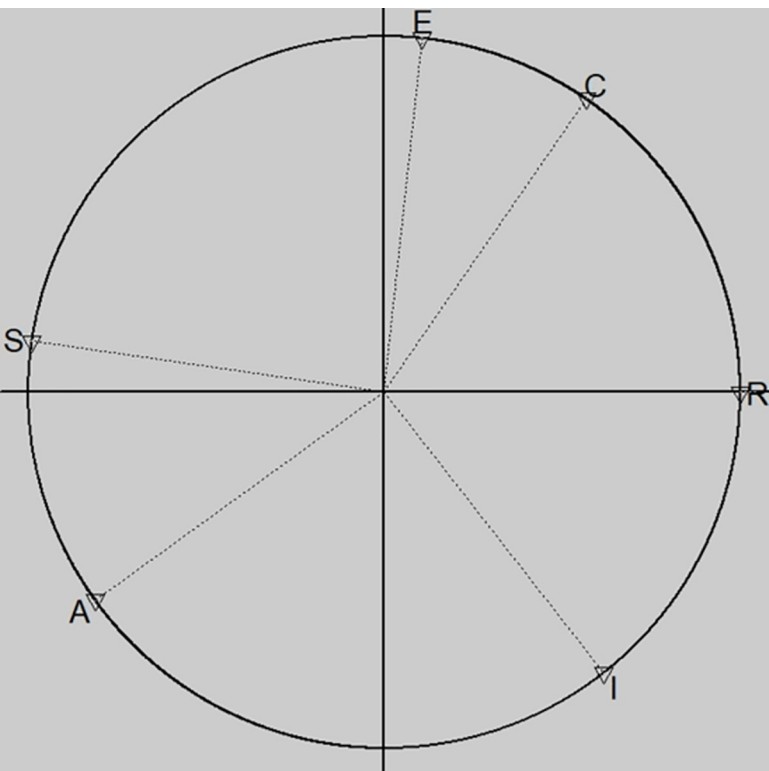

**Fig 2. Data circumplex goodness of fit.**

contrast, young women scored higher on the Artistic and Social dimensions compared to young men. In terms of effect size (Cohen's $d$), the two largest differences were found for Realistic and Social dimensions with an effect size above 0.5, indicating a large-sized difference. Artistic and Conventional dimensions are quite high too compared to Investigative and Enterprising dimensions, with the latter effect not being significant.

**DIF test.** We performed a differential item functioning (DIF) test to examine whether the possible observed gender differences in terms of RIASEC scores can be attributed to specific item bias. The test highlights the number of items in each dimension showing a difference between the two gender groups. Table 6 reports the average DIF score M(β) and which group is favoured ([33]). Results were obtained from the use of the DifR package in R.

Overall, DIF was detected in 54% of all items ($n = 82$), with a majority of 96% in favour of the male group ($n = 79$). Also, the average DIF score for items favouring the male group is higher than that for the items favouring the female group. Important to note, none of the average DIF scores has values greater than 0.2, the usual cut-off point to signal that item bias could affect the responses of one or both gender groups. As such, we have not found any evidence that the group differences in RIASEC scores could be attributed to specific item bias.

**Table 4. Randomization test of hypothesized order relations.**

| Group | Predictions | | Correspondence Index | *p*-value |
|---|---|---|---|---|
| | **Met** | **Tied** | | |
| **Total** | 60 | 0 | 0.67 | 0.02 |
| **Young men** | 57 | 0 | 0.58 | 0.02 |
| **Young women** | 57 | 0 | 0.58 | 0.05 |

**Table 5. Gender differences in interests: Mean, standard deviation, *t*-test and Cohen's *d*.**

|  | Young men | Young women | *t*(1279) | *d* |
|---|---|---|---|---|
| **R** | 33.44 (24.62)[a] | 13.63 (16.05) | 17.33*** | 0.95 |
| **I** | 26.66 (21.59) | 23.78 (19.23) | 2.52* | 0.14 |
| **A** | 28.36 (21.70) | 34.64 (22.99) | -4.99*** | - 0.28 |
| **S** | 24.17 (21.72) | 41.62 (23.04) | -13.84*** | - 0.78 |
| **E** | 27.83 (25.23) | 27.34 (22.71) | 0.36 | 0.02 |
| **C** | 23.63 (23.07) | 19.38 (19.21) | 3.60*** | 0.20 |

Notes:

[a]: Mean (standard deviation); *t*: Student's *t*; *d*: Cohen's *d*

* $p < .05$

** $p < .01$ and

*** $p < .001$. We use [32]'s guidelines to assess the effect size. We say that there is a large effect when $d = 0.5$. In this case, the effect explains 25% of the total variance. There is a medium effect for $d = 0.3$ (the effect explains 9% of the total variance) and a small effect for $d = 0.1$ (1%).

## Gender differences in interest profile diversity

In this section, we focus on two vocational interest profiles: one-dimensional interest types (e.g., R) and two-dimensional interest types (e.g., RI and IR are equivalent). We evaluate their respective diversity from their absolute frequencies among participants' profiles (i.e., richness) and their relative frequencies (i.e., evenness). Table 7 summarises our results.

For one-dimensional interest types, young men and women's profiles have the same richness: each group includes six different vocational interest profiles. However, there is substantially greater evenness among young men's profiles (0.82) than young women's profiles (0.58). The former are more evenly distributed among the six dimensions than the latter (the exact relative frequency in each type is reported in Table 8a in Appendix C). It confirms the greater diversity among young men's one-dimensional interest types, noticeable by comparing Figs 3 and 4. For young women, the R and C dimensions occurred in less than 5% of cases, where all dimensions occurred in more than 5% for young men.

Similar results are obtained in the case of the two-dimensional interest types: the distribution is more even for young men's vocational interest profiles than for those of young women (the exact relative frequencies are gathered in Table 8b in Appendix C). Low frequency types are again more common for young women than young men, noticeable again by comparing Figs 5 and 6. Indeed, for the former, nine two-dimension types each accounted for less than 5% of cases, where this was true for only five two-dimension types for the latter.

**Table 6. Number and percentage of items showing DIF.**

| Dimension | *N* | N items showing DIF | % items showing DIF | Favour young women | | Favour young men | |
|---|---|---|---|---|---|---|---|
| | | | | **n** | **M(β)** | **n** | **M(β)** |
| **R** | 22 | 14 | 64 | 1 | 0.09 | 13 | 0.13 |
| **I** | 29 | 18 | 62 | 0 | - | 18 | 0.14 |
| **A** | 26 | 15 | 58 | 0 | - | 15 | 0.12 |
| **S** | 28 | 17 | 61 | 1 | 0.10 | 16 | 0.12 |
| **E** | 24 | 11 | 46 | 1 | 0.06 | 10 | 0.12 |
| **C** | 23 | 7 | 30 | 0 | | 7 | 0.11 |

**Table 7. Richness and evenness of vocational interests according to gender.**

| | | 704 young women | 577 young men |
|---|---|---|---|
| **One-dimensional interest type (e.g., R)** | **Richness[a]** | 6 | 6 |
| | **Evenness[b]** | 0.58 | 0.82 |
| **Two-dimensional interest type (e.g., RI and IR are equivalent)** | **Richness** | 15 | 15 |
| | **Evenness** | 0.40 | 0.72 |

Notes:

[a]: Richness is equal to the number of vocational interest profiles (cf. column 1 of Table 8a in Appendix C).

[b]: Evenness is calculated from the inverse of the product of the Simpson index (cf. last row of the Table 8a in Appendix C) and the richness value.

## Discussion

The aim of our paper was to document the vocational interests of Scottish adolescents. To do so we transposed a well-established Flemish interest inventory instrument to a Scottish context. We showed that gender differences do exist in vocational interests in our sample of pupils, in line with the classic work of [5] in a RIASEC framework or [16] in an alternative trait framework. We can discuss our findings with respect to three aspects.

First, we showed the successful cross-cultural replication of the Flemish interest inventory in a Scottish context. Indeed, we confirmed the structural validity of this tool by examining the circumplex structure of the Scottish data using both CFA and RTOR. We were able to show an adequate fit of these data with the circular ordering for all samples, in particular for young men. Our three spatial representations corroborated the theoretical RIASEC ordering

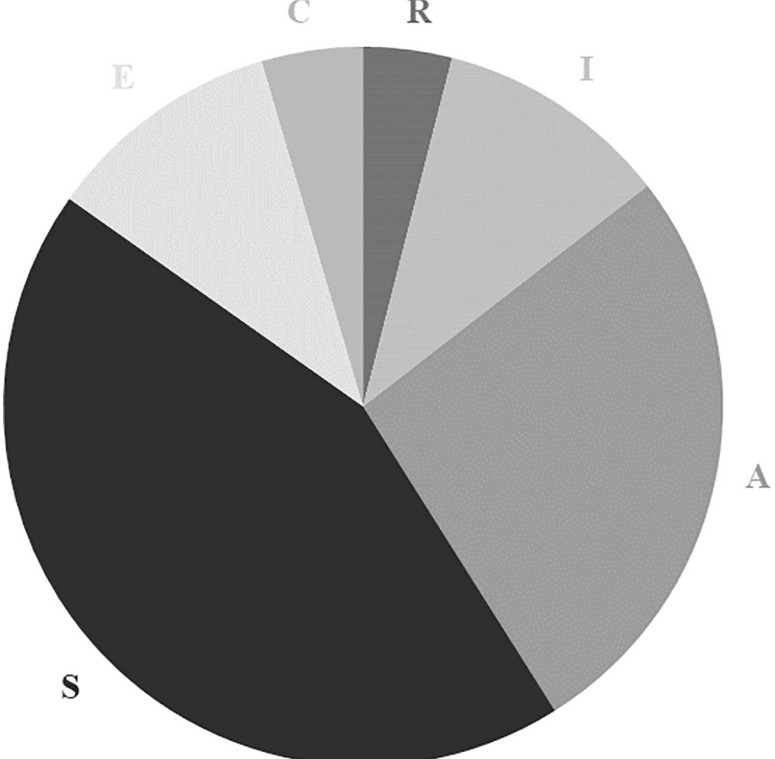

**Fig 3. Distribution of one-dimensional interest types among young women.**

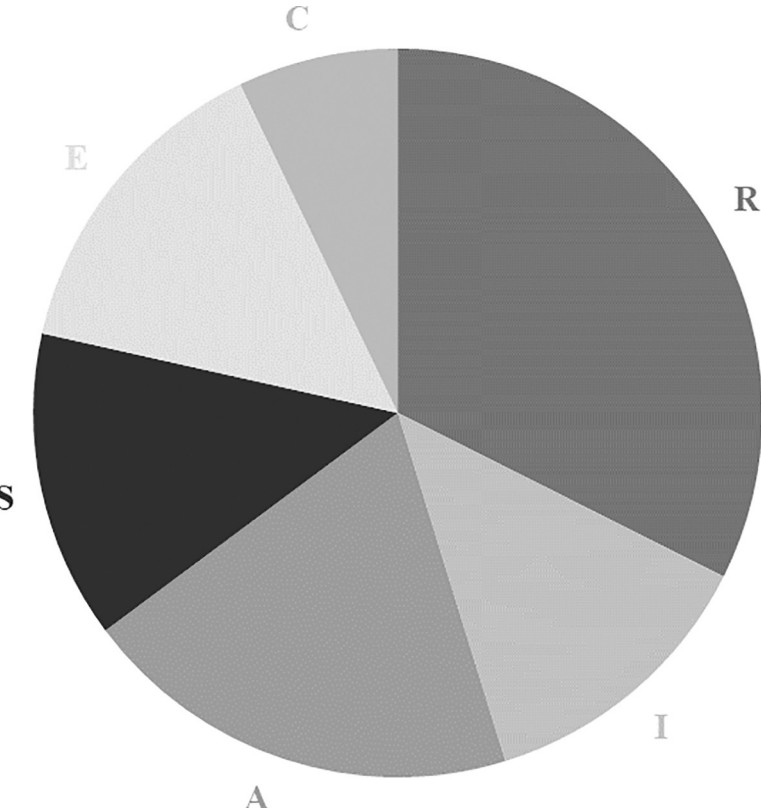

**Fig 4. Distribution of one-dimensional interest types among young men.**

representation. Our results are mostly in line with those of the Flemish benchmark (e.g., RMSEA values were also greater than 0.08 for the young women sample and the total sample, and all SRMR values were less than 0.08), especially considering the younger age of the pupils involved (as suggested by [30] and [31]).

Second, our findings regarding the gender differences in terms of interest scores can be compared with those reported by [11] and [5]. Similar to both studies, the average scores of Scottish young men on the Realistic dimension and those of Scottish young women on the Social and Artistic dimensions were higher than those of their counterparts. Contrary to [11] but similar to [5], we did not find gender differences on the Enterprising dimension. In contrast to both studies, we found significant differences on the Conventional dimension, male pupils scoring higher. Remarkably, at this stage of our study and at odds with [11], the differential item functioning test results imply gender-neutral interest dimensions.

Third, our findings regarding the gender differences in terms of interest profiles are two-fold. On the one hand, they confirm that young women and men's vocational interest profiles are not evenly distributed. Some profiles are more predominant than others. On the other hand, this uneven distribution holds for both the six one-dimensional interest types and the 15 two-dimensional interest types. In the case of the one-dimensional interest type, the proportion of Scots young men with Realistic interest type and that of Scots young women with Social interest type were higher than those of their counterparts. This uneven distribution has been found in other contexts, in particular the US and Swiss labour markets ([7,8]). Note that the relative frequency of each one-dimensional type among the respondents' vocational interest type impacts the relative frequency of each two-dimensional interest type. For instance, we can

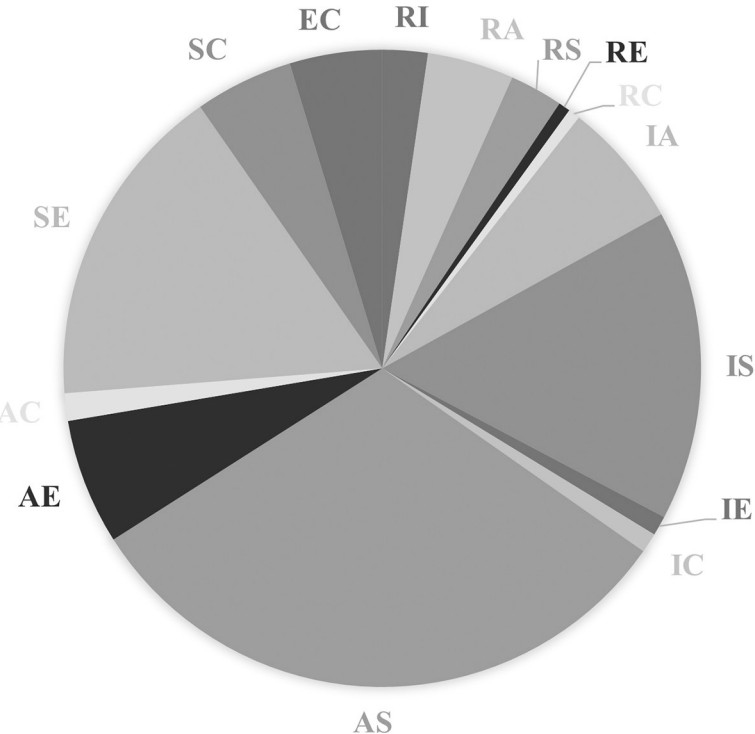

**Fig 5. Distribution of two-dimensional interest types among young women.** Note: The figure is visually simplified by including one combination of letter orderings. For instance, RI includes individuals whose primary interests were ranked R and then I, as well as those who ranked I and then R.

observe the dominance of the two-dimensional interest types combined with Realistic in the case of young men's profiles. This finding contributes to the literature highlighting the greater male variability (or diversity) on several traits.

These findings could signal the environment the participants would prefer to study in and/ or work in. Recall that vocational interests were mapped by Holland to jobs and to academic disciplines by [13] and [15]. Academic disciplines like 'Engineering' or 'Marine science' are associated with R environments, 'Humanities' or 'Social sciences' with S environments. As people tend to choose environments close to their vocational interests and these latter tend to be relatively stable during the transition from adolescence to adulthood ([4,5]), this could indicate their career or the subjects they may wish to study. Observed gender dissimilarities in terms of vocational interest profiles could lead to potential gender differences in terms of educational pathways and career-decision making.

The DIF results leads us to be cautious in that respect. The question of gender fairness in instruments was raised by [34]. Our instrument does not show bias in a Scottish context. Our findings 'signal' the environment the participants would prefer to study in and/or work in. In addition, some pupils may not end up in the artistic environment they favour, as creative industries are known to have high levels of unemployment. In recent years, in Scotland, there have been various campaigns to promote gender balance in STEM subjects and professions with contrasting results. The education sector and the employers are working together to attract more young women in these disciplines, but it is taking time to break stereotypes significantly. More information about the interest types of pupils and exploring the adjacent dimensions in the circular structure could accelerate the process as long as environments are also mapped according

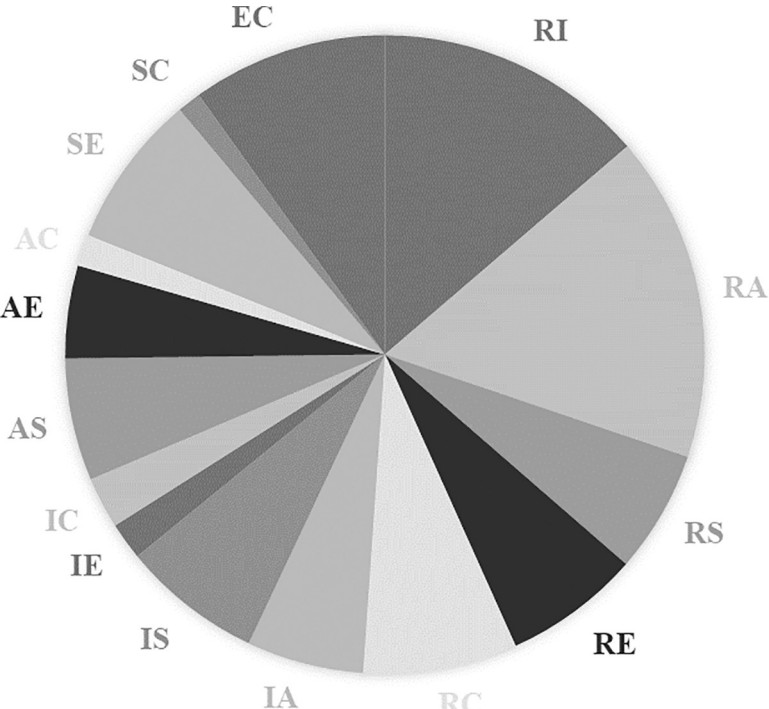

**Fig 6. Distribution of two-dimensional interest types among young men.** Note: The figure is also visually simplified by including one combination of letter orderings.

to similar vocational interest profiles. Scotland is not the only country to attempt to promote gender-parity and it would be worth complementing our documentation with practical suggestions on when/why efforts to achieve this parity may succeed as elegantly elaborated by [19].

## Conclusion

Our paper provides the first documentation of Scottish adolescents' vocational interests. We used Holland's theory of vocational interests and successfully applied a Flemish instrument to a Scottish context. We pointed out (1) the uneven distribution of vocational interests across the participants and (2) the gender differences. These results also provide relevant information to guidance teachers and policy-makers. Guidance teachers could offer additional information, advice and support to young people in their subsequent educational and career paths based on their pupils' vocational interests. Policy-makers could take into account these vocational interests when they design policies aimed at reducing gender disparities in education and labour markets. In Scotland, the task to map education and labour markets in terms of vocational interests forms a promising new research line.

## Appendices

### Appendix A: Simpson index and evenness

We denote by W the richness. The Simpson index λ measures the probability that two participants randomly selected from our sample of young women (or young men) have the same vocational interest type. It is calculated as $\lambda = \sum_{i=1}^{W} p_i^2$ where $p$ is the proportion of the young women (men) from the dataset that have the $i^{\text{th}}$ vocational interest type. Evenness $V$ is calculated as $V = \frac{1}{\lambda W}$.

## Appendix B: Our tool inspired by Ghent University (SIMON-I)

Our tool mirrors some essential elements of a portal designed by the University of Ghent in Belgium and presented in [11]. It aims to help pupils to make study choices and to facilitate research into the relationships between interest types and pupils' choice of academic subjects.

It explores the links between academic subjects that young people may choose to study at University (e.g., Economics, Divinity, Computer Science), activities that they may enjoy or are willing to try (e.g., collecting data, giving travel advice) and occupations that they would like to do (e.g., architect, nurse).

The questionnaire of our tool is almost identical to the survey developed by [11]. Participants responded in a '*Like/Don't like/Prefer not to answer*' format to a series of statements regarding 98 activities and 65 occupations. In our tool, the '*Prefer not to answer*' option was pre-populated to facilitate the completion of the questionnaire. The statement of one activity was amended. 'Reading English language scientific articles' became 'Reading scientific articles'. By dropping the occupation: 'Industrial designer', all 'designer' occupations were Artistic occupations in our case. The mapping of all statements to interest dimensions is identical to [11].

## Appendix C

**Table 8a. Six one-dimensional interest types.**

| Vocational interest profiles | 704 young women | | 577 young men | |
|:---:|:---:|:---:|:---:|:---:|
| | p | p$^2$ | p | p$^2$ |
| R | 0.0398 | 0.0398 | 0.3241 | 0.1050 |
| I | 0.1051 | 0.1051 | 0.1282 | 0.0164 |
| A | 0.2656 | 0.2656 | 0.1958 | 0.0384 |
| S | 0.4375 | 0.4375 | 0.1369 | 0.0187 |
| E | 0.1065 | 0.1065 | 0.1438 | 0.0207 |
| C | 0.0455 | 0.0455 | 0.0711 | 0.0050 |
| *Total* | 1.0000 | 0.2880 *(Simpson Lambda)* | 1.0000 | 0.2043 *(Simpson Lambda)* |

Note: p: Relative frequency.e: p: Relative frequency.

**Table 8b. 15 two-dimensional interest types.**

| Vocational interest profiles | 704 young women | | 577 young men | |
|:---:|:---:|:---:|:---:|:---:|
| | p | p$^2$ | p | p$^2$ |
| RI or IR | 0.023 | 0.001 | 0.137 | 0.019 |
| RA or AR | 0.044 | 0.002 | 0.165 | 0.027 |
| RS or SR | 0.027 | 0.001 | 0.062 | 0.004 |
| RE or ER | 0.006 | 0.000 | 0.069 | 0.005 |
| RC or CR | 0.006 | 0.000 | 0.078 | 0.006 |
| IA or AI | 0.064 | 0.004 | 0.059 | 0.003 |
| IS or SI | 0.158 | 0.025 | 0.071 | 0.005 |
| IE or EI | 0.010 | 0.000 | 0.019 | 0.000 |
| IC or CI | 0.010 | 0.000 | 0.026 | 0.001 |
| AS or SA | 0.313 | 0.098 | 0.062 | 0.004 |
| AE or EA | 0.064 | 0.004 | 0.047 | 0.002 |
| AC or CA | 0.014 | 0.000 | 0.016 | 0.000 |

*(Continued)*

**Table 8b.** (Continued)

| Vocational interest profiles | 704 young women | | 577 young men | |
|---|---|---|---|---|
| | p | p² | p | p² |
| SE or ES | 0.166 | 0.028 | 0.078 | 0.006 |
| SC or CS | 0.050 | 0.002 | 0.012 | 0.000 |
| EC or CE | 0.047 | 0.002 | 0.099 | 0.010 |
| *Total* | 1.0000 | 0.1666 (λ) | 1.0000 | 0.0925 (λ) |

Note: p: Relative frequency.

## Acknowledgments

We are very grateful to the staff and pupils who participated in this work in the following secondary schools: Auchenharvie Academy, Auchmuty High School, Brae High School, Castlebay Community School, Dingwall Academy, Dunoon Grammar School, Hermitage Academy, Islay High School, Kirkcaldy High School, Kirkwall Grammar School, Levenmouth Academy, Lochgelly High School, Lomond School, Madras College, Oban High School, St Andrew's RC High School, Stromness Academy and Wellington School. All errors are ours. The views expressed in this paper are ours and do not represent those of these schools or our Universities.

## Author Contributions

**Conceptualization:** Laurence Lasselle, Stijn Schelfhout, Lot Fonteyne, Graham Kirby, Ian Smith, Wouter Duyck.

**Data curation:** Laurence Lasselle, Graham Kirby.

**Formal analysis:** Laurence Lasselle, Stijn Schelfhout, Lot Fonteyne.

**Investigation:** Laurence Lasselle, Stijn Schelfhout, Lot Fonteyne.

**Methodology:** Laurence Lasselle, Stijn Schelfhout, Lot Fonteyne.

**Project administration:** Laurence Lasselle, Graham Kirby.

**Software:** Lot Fonteyne, Graham Kirby.

**Validation:** Stijn Schelfhout, Lot Fonteyne.

**Visualization:** Laurence Lasselle, Stijn Schelfhout.

**Writing – original draft:** Laurence Lasselle, Stijn Schelfhout.

**Writing – review & editing:** Laurence Lasselle, Stijn Schelfhout, Lot Fonteyne, Graham Kirby, Ian Smith, Wouter Duyck.

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
