## [Decision Letter · Decision Letter 0]

21 Jun 2021

PONE-D-21-04302

An Examination of Gender Imbalance in Scottish Adolescents' Vocational Interests

PLOS ONE

Dear Dr. Lasselle,

Thank you for submitting your manuscript to PLOS ONE. After careful consideration, we feel that it has merit but does not fully meet PLOS ONE’s publication criteria (esp. no. 3 and 4) as it currently stands. Therefore, we invite you to submit a revised version of the manuscript that addresses the points raised during the review process but it is not automatically necessary to implement all the suggestions.

We look forward to receiving your revised manuscript.

Kind regards,

Frantisek Sudzina

Academic Editor

PLOS ONE

Journal Requirements:

2. Please change "female” or "male" to "woman” or "man" or other as appropriate, when used as a noun (see for instance https://apastyle.apa.org/style-grammar-guidelines/bias-free-language/gender).

3. Please provide additional details regarding participant consent. In the ethics statement in the Methods and online submission information, please ensure that you have specified whether your IRB specifically approved the consent procedure and the opt-out procedure for parental consent. Thank you for your attention to this request.

4. Please include your tables as part of your main manuscript and remove the individual files. Please note that supplementary tables should be uploaded as separate "supporting information" files.

Reviewers' comments:

Reviewer's Responses to Questions

**Comments to the Author**

1. Is the manuscript technically sound, and do the data support the conclusions?

Reviewer #1: Yes

Reviewer #2: No

2. Has the statistical analysis been performed appropriately and rigorously? 

Reviewer #1: Yes

Reviewer #2: No

3. Have the authors made all data underlying the findings in their manuscript fully available?

Reviewer #1: No

Reviewer #2: Yes

4. Is the manuscript presented in an intelligible fashion and written in standard English?

Reviewer #1: Yes

Reviewer #2: Yes

5. Review Comments to the Author

Reviewer #1: In PONE-D-21-04302, the authors seek to validate the RIASEC model of vocational interests in a large sample of Scottish adolescents, as well as characterize sex differences in vocational interests. The sample is large, and the study will be of interest to those who follow vocational interests, and sex differences more generally. I think the manuscript is generally well written, and should be acceptable for publication following some (relatively) minor revisions. Below I outline a few specific considerations, and suggestions. My primary concerns are rooting the present work a bit more in a theoretical context regarding sex differences more generally (for a fabulous recent summary of this area, see Archer, 2019: https://onlinelibrary.wiley.com/doi/abs/10.1111/brv.12507), as well as some refinement of the text itself. I hope the authors find these comments useful, as this manuscript was a well prepared and enjoyable read.

1) Page 4. There is an unclear referent when the authors mention “this seminal documentation.” This could be easily remedied by altering the text to “Holland’s seminal documentation…”

2) Page 5/6. The authors mention “richness and evenness.” This seems to be tied to related work about what is often referred to as the “Male Variability Hypothesis,” common in evolutionary biology and evolutionary psychology. The present study could be firmly rooted in this theoretical framework. A recent paper about sex differences in STEM (https://journals.sagepub.com/doi/full/10.1177/0890207020962326) offers a nice summary of the relevant considerations. Briefly adding this framework would move the introduction beyond simpler description, and complement this description an established theoretical framework that specifically predicts greater male variability on a number of traits (vocational interest being one possibility).

3) Page 6. The authors employ sex-related terms (male/female) when discussing gender (e.g., man/woman or boy/girl). As sex and gender are my primary area of research expertise, I am well aware that the vast majority of individuals have sex that is consistent with their gender (i.e., most males are men, and most females identify as women). That said, a great number of people ruthlessly critique the mixing of sex and gender terminology, so it might be helpful to be as precise as possible here in specifying whether sex (male/female) or gender (boy/girl or man/woman) was measured, and keep the terminology consistent throughout. I understand that this is perhaps a frustrating request, one I myself have been irritated by in my own field. One possible solution is to simply add an additional footnote to this sentence that reads: “The authors are aware that some individuals’ gender does not align with their biological sex. Because the majority of individuals nonetheless express gender that accords with their sex (e.g., Zucker, 2017: https://www.publish.csiro.au/sh/SH17067), we employ gender terminology throughout.” This same consideration applies to page 7. Did students self-declare their sex (male/female), or their gender (boy/girl)? Again, I know that most people use these terms interchangeably, and I certainly understand what the authors mean to convey. However, one can never be too careful.

4) Page 11. It isn’t entirely clear to me what the authors mean when they state: “The heigh determines the type.” This might be more obvious to readers who are highly familiar with the RIASEC model, but a brief explanation (or alternative wording) might help those who are less initiated.

5) Pages 12-15. It strikes me as somewhat odd to have a rather lengthy bullet-point list of procedures. Many of these procedural minutiae are not particularly relevant to a reader’s understanding of how the data were collected. A brief paragraph describing the procedures would suffice, and also reduce the manuscript text a bit.

6) Page 17. The authors mention the RANDALL package. Could the authors clarify at some point in the text which statistical software was used for the analysis, so curious readers can themselves follow-up more easily by finding the appropriate approach and analytic packages? Similarly, it is not entirely clear to me what RTOR analysis is. This section just needs a bit more contextualization, in my opinion. (The authors are more specific on page 18, when they mention the DifR package in R.)

7) Table 5 (mention of sex differences on Page 18. Both differences mentioned appear to be closer to Cohen’s benchmarks for large differences (i.e., d ≥0.80). The authors seem to be understating this difference somewhat.

8) The authors might consider citing one of the largest cross-cultural studies of occupation preferences that has been undertaken (https://link.springer.com/article/10.1007/s10508-008-9380-7). This study also found consistent sex differences that complement the present findings. Although Lippa (2010) did not use the RIASEC, the conclusions are nonetheless complementary. The Lippa (2010) study stands as a classic in this area just in much the same way that the Su et al. (2009) study does.

9) Page 22. For readers whose memories are not always operating at peak efficiency (like myself), it might be helpful if the authors mentioned once again what the acronym DIF stands for. In the paragraph that follows, I think the Stewart-Williams & Halsey (2021) reference that I mentioned above would be a natural complement to this paragraph. Many attempts to bring gender-parity to certain vocations are likely doomed to failure, although these authors offer some practical suggestions on when/why such efforts might succeed.

10) Figure 4a and 4b. This is entirely a stylistic choice, but the figure itself might be visually simplified by only including one combination of letter orderings, and including a note on the figure that each order (e.g., EC) includes individuals whose primary interests were ranked E and then C, as well as those who ranked C and then E.

11) Table 3. Could the authors include exact p-values for each model as well? This is important information to include while readers decide for themselves whether the models accurately capture the structure of the data.

Reviewer #2: PONE-D-21-04302

An Examination of Gender Imbalance in Scottish Adolescents' Vocational Interests

The manuscript is a descriptive study of RIASEC vocational interests among a sample of 1,306 Scottish students from 18 secondary schools. The study evaluates the structural validity, and the gendered distribution of the RIASEC scales items with a measure (SIMON-I) developed by Fonteyne et al. (2017) to assess students in the Flemish education system. In essence, the paper is a cross-cultural validation study of the SIMON-I.

Given that there is a relatively large literature in vocational psychology on the structural validity and gender distribution of Holland’s (1997) RIASEC personality types across cultural (country) samples, the issue with the present manuscript is how does this study contribute. There seems to be two contributions 1) the Forteyne et al. measure is relatively new and has to my knowledge not been evaluated in other countries; and 2) the RIASEC model has yet to be evaluated with a sample of Scottish students. These contributions fit the ongoing discussion of Holland’s model but do not add a variation on the model, extend the methodology of testing circular models, and add a new way to conceptualize interest models and cross-cultural assessment of Holland’s model. The gender differences among Scottish students are similar to findings from Su’s et al. (2009) meta-analysis of US samples. Recent research on the circular model have extended their studies to multiple cultures (nations)—see Glosenberg et al. (2019).

There are several methodological issues that can be addressed, beginning with the interpretation of the CFA.

1. The authors state on p. 16 that “Although most index values are of at least acceptable quality, the RMSEA values are too high.” They then dismiss the RMSEA as a fit measure because of degrees of freedom being small. The dismissal is premature, see Nagy et al. (2011) Table 3 and Table 4 and their explanation of the CFA for 7 different Holland models. Clearly, the CFA model tested by the current authors does not fit. I would add that it would strengthen the paper to test more than one variation of Holland’s model.

2. The footnote on page is inappropriate because you do not apply significant tests to the RIASEC intercorrelations since they are not independent. That is the reason the randomization test was introduced (Hubert & Arabie, 1987).

3. Interpretation of benchmarks and the p-value of the randomization test is incorrect. On page 17 the authors state, “All our correspondence indices (CI) are well above the international benchmarks (0.48) suggested by Rounds and Tracey (1996), while a circular fit of the data reached a significance of at least p = 0.05.” Rounds and Tracey reported that the mean CI was .48 for international matrices, almost 3 SDs below US mean CI. Most of the international samples did not support a circular structure. The correct interpretation of the p-value is: Given the circular model of RIASEC, statistically significant p-value indicates that random ordering of the RIASEC types can be rejected.

4. Over interpretation of the structural tests. On page 18, the authors state, “In summary, the circular fit for our data has been confirmed by both CFA and RTOR analyses. As a consequence, the use of the Flemish instrument is suited for cross-cultural application to a Scottish context.” Structural tests are necessary but not sufficient for application of measures cross-culturally.

Alexander Glosenberg, Terence J.G. Tracey, Tara S. Behrend, David L. Blustein, Lori L. Foster, (2019). Person-vocation fit across the world of work: Evaluating the generalizability of the circular model of vocational interests and social cognitive career theory across 74 countries,

Journal of Vocational Behavior, 112, 92-108.

Hubert, L., & Arabie, P. (1987). Evaluating order hypotheses with proximity matrices. Psychological Bulletin, 102, 172-178.

Nagy, G., U. Trautwein, U., & Lüdtke, O. (2010). The structure of vocational interests in Germany: Different methodologies, different conclusions. Journal of Vocational Behavior, pp. 153-169.

6. PLOS authors have the option to publish the peer review history of their article (what does this mean?). If published, this will include your full peer review and any attached files.

Reviewer #1: **Yes: **Scott W. Semenyna

Reviewer #2: No

---

## [Author Response · Author response to Decision Letter 0]

5 Sep 2021

 We trust that our manuscript meets these requirements. 

2. Please change "female” or "male" to "woman” or "man" or other as appropriate, when used as a noun (see for instance https://apastyle.apa.org/style-grammar-guidelines/bias-free-language/gender).

We have made the change. 

3. Please provide additional details regarding participant consent. In the ethics statement in the Methods and online submission information, please ensure that you have specified whether your IRB specifically approved the consent procedure and the opt-out procedure for parental consent. Thank you for your attention to this request.

We have revised our ethics statement in the light of your request. We have clearly specified that our research was carried out in accordance with the requirements of the institution ethics committee regarding online informed consent, the opt-out procedure for parental consent and data availability. Note that we have added details regarding participant consent. This is indeed particularly important as the age of full legal capacity is 16 in Scotland. 

4. Please include your tables as part of your main manuscript and remove the individual files. Please note that supplementary tables should be uploaded as separate "supporting information" files.

All tables have been included as part of our manuscript. 

We have revised our Data Availability statement. Our statement reflects the requirements of our IRB.

 Our manuscript has been amended.

 

Reviewers' comments:

Reviewer's Responses to Questions

Comments to the Author

1. Is the manuscript technically sound, and do the data support the conclusions?

Reviewer #1: Yes

Reviewer #2: No

We wish to draw to the attention of Reviewer #2 that we have considered their comments and revised our research methods, results and discussion sections.

2. Has the statistical analysis been performed appropriately and rigorously? 

Reviewer #1: Yes

Reviewer #2: No

We wish to draw to the attention of Reviewer #2 that we have considered their comments and revised our research methods and results sections.

3. Have the authors made all data underlying the findings in their manuscript fully available?

Reviewer #1: No

Reviewer #2: Yes

Our research was carried out in accordance with the requirements of our IRB regarding data availability. Namely, the individual responses of our participants to the RIASEC test are not fully available. However, all relevant summary statistics are included in our manuscript. 

4. Is the manuscript presented in an intelligible fashion and written in standard English?

Reviewer #1: Yes

Reviewer #2: Yes

5. Review Comments to the Author

 

Reviewer #1: In PONE-D-21-04302, the authors seek to validate the RIASEC model of vocational interests in a large sample of Scottish adolescents, as well as characterize sex differences in vocational interests. The sample is large, and the study will be of interest to those who follow vocational interests, and sex differences more generally. I think the manuscript is generally well written, and should be acceptable for publication following some (relatively) minor revisions. Below I outline a few specific considerations, and suggestions. My primary concerns are rooting the present work a bit more in a theoretical context regarding sex differences more generally (for a fabulous recent summary of this area, see Archer, 2019: https://onlinelibrary.wiley.com/doi/abs/10.1111/brv.12507), as well as some refinement of the text itself. I hope the authors find these comments useful, as this manuscript was a well prepared and enjoyable read.

We are very grateful for your detailed comments which have greatly improved our paper. 

In the light of your feedback, we have revised all sections of our manuscript, with particular attention to the theoretical context. Thank you very much for your constructive comments again. 

1) Page 4. There is an unclear referent when the authors mention “this seminal documentation.” This could be easily remedied by altering the text to “Holland’s seminal documentation…”

We have amended the text.

To facilitate the discussion, we limit the presentation of this first documentation to Holland’s one- and two-dimensional interest types. 

2) Page 5/6. The authors mention “richness and evenness.” This seems to be tied to related work about what is often referred to as the “Male Variability Hypothesis,” common in evolutionary biology and evolutionary psychology. The present study could be firmly rooted in this theoretical framework. A recent paper about sex differences in STEM (https://journals.sagepub.com/doi/full/10.1177/0890207020962326) offers a nice summary of the relevant considerations. Briefly adding this framework would move the introduction beyond simpler description, and complement this description an established theoretical framework that specifically predicts greater male variability on a number of traits (vocational interest being one possibility).

We have revised and significantly extended our introduction and background sections. We trust that our manuscript has now adequate theoretical context regarding sex differences. 

3) Page 6. The authors employ sex-related terms (male/female) when discussing gender (e.g., man/woman or boy/girl). As sex and gender are my primary area of research expertise, I am well aware that the vast majority of individuals have sex that is consistent with their gender (i.e., most males are men, and most females identify as women). That said, a great number of people ruthlessly critique the mixing of sex and gender terminology, so it might be helpful to be as precise as possible here in specifying whether sex (male/female) or gender (boy/girl or man/woman) was measured, and keep the terminology consistent throughout. I understand that this is perhaps a frustrating request, one I myself have been irritated by in my own field. One possible solution is to simply add an additional footnote to this sentence that reads: “The authors are aware that some individuals’ gender does not align with their biological sex. Because the majority of individuals nonetheless express gender that accords with their sex (e.g., Zucker, 2017: https://www.publish.csiro.au/sh/SH17067), we employ gender terminology throughout.” This same consideration applies to page 7. Did students self-declare their sex (male/female), or their gender (boy/girl)? Again, I know that most people use these terms interchangeably, and I certainly understand what the authors mean to convey. However, one can never be too careful.

Thank you very much for drawing our attention to this point. We trust that our manuscript follows the APA guidelines. 

4) Page 11. It isn’t entirely clear to me what the authors mean when they state: “The heigh determines the type.” This might be more obvious to readers who are highly familiar with the RIASEC model, but a brief explanation (or alternative wording) might help those who are less initiated.

Page 13 (paper with track changes) / Page 12 (paper without track changes). 

We have revised the sentence. By replacing the word ‘Height’ by ‘score’, we trust that the sentence is clearer. Score is now consistently used throughout the manuscript. It is first mentioned in page 4 (currently 5 – version with track changes). 

With regards to this specific page, note that the example that follows in the text uses the word ‘score’. 

5) Pages 12-15. It strikes me as somewhat odd to have a rather lengthy bullet-point list of procedures. Many of these procedural minutiae are not particularly relevant to a reader’s understanding of how the data were collected. A brief paragraph describing the procedures would suffice, and also reduce the manuscript text a bit.

Page 14 (paper with track changes) / Page 12 (paper without track changes).

We have entirely revised our ethics statement. The section has been shortened and is a page long. 

6) Page 17. The authors mention the RANDALL package. Could the authors clarify at some point in the text which statistical software was used for the analysis, so curious readers can themselves follow-up more easily by finding the appropriate approach and analytic packages? Similarly, it is not entirely clear to me what RTOR analysis is. This section just needs a bit more contextualization, in my opinion. (The authors are more specific on page 18, when they mention the DifR package in R.)

This section of our manuscript has now been revised in the light of our comment.

CirCe and RANDALL package: Page 20 (paper with track change) and page 15 (paper without track changes). 

For the reader’s convenience, the manuscript also provides a more elaborate description of the RTOR analyses (from page 10 - paper with track changes and page 9 - paper without track changes).

7) Table 5 (mention of sex differences on Page 18. Both differences mentioned appear to be closer to Cohen’s benchmarks for large differences (i.e., d ≥0.80). The authors seem to be understating this difference somewhat.

The text matches the comments of Table 5 notes. 

8) The authors might consider citing one of the largest cross-cultural studies of occupation preferences that has been undertaken (). This study also found consistent sex differences that complement the present findings. Although Lippa (2010) did not use the RIASEC, the conclusions are nonetheless complementary. The Lippa (2010) study stands as a classic in this area just in much the same way that the Su et al. (2009) study does.

We have now included the reference in the background and discussion sections. Please note that our manuscript also refers to cross-cultural studies in our revised background section.

9) Page 22. For readers whose memories are not always operating at peak efficiency (like myself), it might be helpful if the authors mentioned once again what the acronym DIF stands for. In the paragraph that follows, I think the Stewart-Williams & Halsey (2021) reference that I mentioned above would be a natural complement to this paragraph. Many attempts to bring gender-parity to certain vocations are likely doomed to failure, although these authors offer some practical suggestions on when/why such efforts might succeed.

We have mentioned again the meaning of the abbreviation and the reference has been added in our revised background and discussion sections. 

10) Figure 4a and 4b. This is entirely a stylistic choice, but the figure itself might be visually simplified by only including one combination of letter orderings, and including a note on the figure that each order (e.g., EC) includes individuals whose primary interests were ranked E and then C, as well as those who ranked C and then E.

Both figures have been visually simplified.

11) Table 3. Could the authors include exact p-values for each model as well? This is important information to include while readers decide for themselves whether the models accurately capture the structure of the data.

Page 20 (paper with track changes) / Page 15 (paper without track changes). 

The manuscript now mentions the p-value for the chi-squared model statistics. 

 

Reviewer #2: PONE-D-21-04302

An Examination of Gender Imbalance in Scottish Adolescents' Vocational Interests

The manuscript is a descriptive study of RIASEC vocational interests among a sample of 1,306 Scottish students from 18 secondary schools. The study evaluates the structural validity, and the gendered distribution of the RIASEC scales items with a measure (SIMON-I) developed by Fonteyne et al. (2017) to assess students in the Flemish education system. In essence, the paper is a cross-cultural validation study of the SIMON-I.

Given that there is a relatively large literature in vocational psychology on the structural validity and gender distribution of Holland’s (1997) RIASEC personality types across cultural (country) samples, the issue with the present manuscript is how does this study contribute. There seems to be two contributions 1) the Forteyne et al. measure is relatively new and has to my knowledge not been evaluated in other countries; and 2) the RIASEC model has yet to be evaluated with a sample of Scottish students. These contributions fit the ongoing discussion of Holland’s model but do not add a variation on the model, extend the methodology of testing circular models, and add a new way to conceptualize interest models and cross-cultural assessment of Holland’s model. The gender differences among Scottish students are similar to findings from Su’s et al. (2009) meta-analysis of US samples. Recent research on the circular model have extended their studies to multiple cultures (nations)—see Glosenberg et al. (2019).

There are several methodological issues that can be addressed, beginning with the interpretation of the CFA.

We are very grateful for your detailed comments which have greatly improved our paper. 

In the light of your feedback, we have revised all sections of our manuscript, with particular attention to the theoretical context and the methodological issues. Thank you very much for your constructive comments. 

1. The authors state on p. 16 that “Although most index values are of at least acceptable quality, the RMSEA values are too high.” They then dismiss the RMSEA as a fit measure because of degrees of freedom being small. The dismissal is premature, see Nagy et al. (2011) Table 3 and Table 4 and their explanation of the CFA for 7 different Holland models. Clearly, the CFA model tested by the current authors does not fit. I would add that it would strengthen the paper to test more than one variation of Holland’s model.

Pages 21-24 – paper with track changes / Pages 16 -18 paper without track changes

We agree that the model testing was not reported as extensively as we could have. 

We have tested several models. The model with equal dimension communalities and varying dimension locations provided the best solution (page 24 – paper with track change / page 18 – paper without track change). We considered reporting all analyses directly. However, we deem such elaborate reports beyond the scope of the present manuscript, as the present study focuses on gender interest differences and not on the debate about structural RIASEC analyses as such. 

However, we do acknowledge that the RIASEC structure of the data remains important and needs to be addressed properly. As such, we have reported both CFA and RTOR as these methods represent the current state of the art. We agree that the RMSEA values are too high and that the models do not reach a good fit as such. However, caution is advised when basing the judgment of (absence of) goodness of fit on the performance of one fit index. A full set of indices in addition to other information can provide a more comprehensive interpretation of model fit:

“Using the RMSEA to assess the model fit in models with small df is problematic and potentially misleading unless the sample size is very large. We urge researchers, reviewers, and editors not to dismiss models with large RMSEA values with small df without examining other information. In fact,

we think that it advisable for researchers to completely avoid computing the RMSEA when model df are small.” (Kenny et al., 2015, p 503)

As such, we started our analyses with a CFA that fits a circumplex model to the full variance-covariance matrix (explanation elaborated in page 10 – paper with track change / page 9 – paper without track changes, result given in page 20 – paper with track change / page 15 – paper without track changes).

Although the other indices indicate at least an adequate fit, the RMSEA values were too high as you correctly indicated. However, this phenomenon also occurred in the original SIMON study (Fonteyne et al., 2017). As the degrees of freedom were quite low, we decided to include other information, as suggested by Kenny and colleagues (2015).

The RTOR did reveal a sufficient circular fit. However, the RTOR evaluates the possibility of a circular structure, but does not evaluate a full fit between variance-covariance matrix of the data and the proposed model (Nagy et al., 2010). Still, for the present study, the empirical evidence towards a circular fit is considered key. As the debate on the most appropriate measure of circular fit is still ongoing (Darcy & Tracey, 2007), we have decided to report both measures so readers can make their own evaluation. On the one hand, we agree with Kenny and colleagues that models should not be dismissed on the results of one (RMSEA) index and that other information should at least be considered. On the other hand, we do want to provide complete information. 

We have also mentioned two studies [Passler & Hell, 2020 and Tracey & Caulum, 2015] that can explain why we observe somewhat mixed results regarding the circular structure of our data (pages 24 and 30 – paper with track changes / pages 18 and 23 – paper without track changes). Indeed, we must note the age of the pupils involved, as the stability (and thus also the stability of the circular structure) of interests increases towards adulthood. 

2. The footnote on page is inappropriate because you do not apply significant tests to the RIASEC intercorrelations since they are not independent. That is the reason the randomization test was introduced (Hubert & Arabie, 1987).

The footnote was inappropriate and has been deleted. 

3. Interpretation of benchmarks and the p-value of the randomization test is incorrect. On page 17 the authors state, “All our correspondence indices (CI) are well above the international benchmarks (0.48) suggested by Rounds and Tracey (1996), while a circular fit of the data reached a significance of at least p = 0.05.” Rounds and Tracey reported that the mean CI was .48 for international matrices, almost 3 SDs below US mean CI. Most of the international samples did not support a circular structure. The correct interpretation of the p-value is: Given the circular model of RIASEC, statistically significant p-value indicates that random ordering of the RIASEC types can be rejected.

We adjusted the exact wordings so that the manuscript now reports a correct interpretation of the p-value (page 23 version with track changes / page 17, version with no track changes). 

4. Over interpretation of the structural tests. On page 18, the authors state, “In summary, the circular fit for our data has been confirmed by both CFA and RTOR analyses. As a consequence, the use of the Flemish instrument is suited for cross-cultural application to a Scottish context.” Structural tests are necessary but not sufficient for application of measures cross-culturally.

Alexander Glosenberg, Terence J.G. Tracey, Tara S. Behrend, David L. Blustein, Lori L. Foster, (2019). Person-vocation fit across the world of work: Evaluating the generalizability of the circular model of vocational interests and social cognitive career theory across 74 countries,

Journal of Vocational Behavior, 112, 92-108.

Hubert, L., & Arabie, P. (1987). Evaluating order hypotheses with proximity matrices. Psychological Bulletin, 102, 172-178.

Nagy, G., U. Trautwein, U., & Lüdtke, O. (2010). The structure of vocational interests in Germany: Different methodologies, different conclusions. Journal of Vocational Behavior, pp. 153-169.

Pages 21-24 – paper with track changes / pages 16-18 – paper without track changes

We have revisited our interpretation of the structural tests. 

Our revised manuscript referred to cross-cultural studies on multiple occasions, including the background section. 

6. PLOS authors have the option to publish the peer review history of their article (what does this mean?). If published, this will include your full peer review and any attached files.

Do you want your identity to be public for this peer review? For information about this choice, including consent withdrawal, please see our Privacy Policy.

Reviewer #1: Yes: Scott W. Semenyna

Reviewer #2: No

 All our figures have been checked by PACE.

---

## [Editor Report · Decision Letter 1]

9 Sep 2021

An Examination of Gender Imbalance in Scottish Adolescents' Vocational Interests

PONE-D-21-04302R1

Dear Dr. Lasselle,

We’re pleased to inform you that your manuscript has been judged scientifically suitable for publication and will be formally accepted for publication once it meets all outstanding technical requirements.

Kind regards,

Frantisek Sudzina

Academic Editor

PLOS ONE
---

## [Editor Report · Acceptance letter]

14 Sep 2021

PONE-D-21-04302R1 

An examination of gender imbalance in Scottish adolescents’ vocational interests 

Dear Dr. Lasselle:

I'm pleased to inform you that your manuscript has been deemed suitable for publication in PLOS ONE. Congratulations! Your manuscript is now with our production department. 

Kind regards, 

on behalf of

Dr. Frantisek Sudzina 

Academic Editor

PLOS ONE